# StratLearner: Learning a Strategy for Misinformation Prevention in Social Networks

**Guangmo Tong**

Department of Computer and Information Sciences

University of Delaware

amotong@udel.edu

## Abstract

*Given a combinatorial optimization problem taking an input, can we learn a strategy to solve it from the examples of input-solution pairs without knowing its objective function?* In this paper, we consider such a setting and study the misinformation prevention problem. Given the examples of attacker-protector pairs, our goal is to learn a strategy to compute protectors against future attackers, without the need of knowing the underlying diffusion model. To this end, we design a structured prediction framework, where the main idea is to parameterize the scoring function using random features constructed through distance functions on randomly sampled subgraphs, which leads to a kernelized scoring function with weights learnable via the large margin method. Evidenced by experiments, our method can produce near-optimal protectors without using any information about the diffusion model, and it outperforms other possible graph-based and learning-based methods by an evident margin.

## 1 Introduction

The online social network has been an indispensable part of today's community, but it is also making misinformation like rumor and fake news widespread [1, 2]. During COVID-19, there have been more than 150 rumors identified by Snopes.com [3]. Misinformation prevention (MP) limits the spread of misinformation by launching a positive cascade, assuming that the users who have received the positive cascade will not be conceived by the misinformation. Such a strategy has been considered as feasible [4], and now fact-checking services are trending on the web, such as Snopes.com [5] and Factcheck.org [6]. Formally, information cascades start to spread from their seed nodes, and the propagation process is governed by an underlying diffusion model. Given the seed nodes (attacker) of the misinformation, the MP problem seeks the seed nodes (protector) of the positive cascade such that the spread of misinformation can be maximally limited.

**MP without Knowing the Diffusion Model.** Existing works often assume that the parameters in the diffusion model are known to us, and they focus primarily on algorithmic analysis for selecting seed nodes [7, 8]. However, the real propagation process is often complicated, and in reality, we can only have certain types of historical data with little to none prior knowledge of the underlying diffusion model. In this paper, we adopt the well-known triggering model [9] to formulate the diffusion process and assume that the parameters are unknown. Now we are given the social graph together with a collection of historical attacker-protector pairs where the protectors were successful, and the goal is to design a learning scheme to compute the best protector against a new attacker. Given the ground set $V$ of the users, the MP problem is given by a mapping $\arg\max_P f(M, P) : 2^V \to 2^V$, where $f$ is the objective function determined by the underlying diffusion model to quantify the prevention effect of the protector $P \subseteq V$ against the attacker $M \subseteq V$. Therefore, our problem is nothing but to learn a mapping from $2^V$ (attacker) to $2^V$ (protector) using training examples $\{(M_i, \arg\max_P f(M_i, P))\}$.

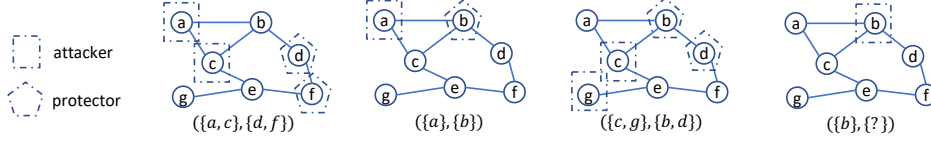

Figure 1: Learning a MP strategy. Suppose that we are given the graph and the information that when the attackers are $\{a,c\}, \{a\}$ and $\{c,g\}$, the best protectors are, respectively, $\{d,f\}, \{b\}$ and $\{b,d\}$. Which is the best protector against the attacker $\{b\}$?

See Fig. 1 for an illustration. While this problem is supervised by the attacker-protector pairs, it is somehow different from the common ones in that it attempts to learn a solution to an optimization problem. One challenge in solving it is that the input and output are sets, while machine learning methods often struggle to deal with objects invariant to permutation [10]. Another challenge lies in properly integrating the graph information into the learning design. As we will see later, directly applying existing methods like graph convolutional networks [11] cannot produce good protectors.

**StratLearner.** We propose a method called StratLearner to solve the considered problem. StratLearner aims to learn a scoring function $f^*(M, S)$ that satisfies

$$f\big(M, \arg\max_P f^*(M, P)\big) \approx \max_P f\big(M, P\big)$$

for each $M$, and if successful, the prediction $\arg\max_P f^*(M, P)$ ensures a good protector. The key idea of StratLearner is to parameterize $f^*(M, P)$ by $f^*(M, P) = \mathbf{w}^\mathsf{T} \mathbf{G}(M, P)$ where $\mathbf{G}(M, P) \in \mathbb{R}^K$ is a feature function constructed through $K \in \mathbb{Z}^+$ random subgraphs with $\mathbf{w} \in \mathbb{R}^K$ being the tunable weights. Our parameterization is justified by the fact that for each distribution over $(M, P)$ and any possible $f$ given by a triggering model, there exists a $\mathbf{w}^\mathsf{T} \mathbf{G}(M, P)$ that can be arbitrarily close to $f$ in the Hilbert space provided that $K$ is sufficiently large. Therefore, StratLearner first generates a collection of random features to obtain $\mathbf{G}(M, P)$, and then learns the weight $\mathbf{w}$ through structural SVM, where a new loss-augmented inference method has been designed to overcome the NP-hardness in computing the exact inference. Our experiments not only show that StratLearner can produce high-quality protectors but also verifies that StratLearner indeed benefits from the proposed feature construction.

## 2 Problem Setting

We proceed by introducing the diffusion model followed by defining the MP problem together with the learning settings.

### 2.1 Model

We consider a social network given by a directed graph $G = (V, E)$. Each node $u \in V$ is associated with a distribution $\mathcal{N}_u(S)$ over $2^{N_u^-}$ with $N_u^-$ being the set of the in-neighbors of $u$; each edge $(u, v) \in E$ is associated with a distribution $\mathcal{T}_{(u,v)}(x)$ over $(0, +\infty)$ denoting the transmission time. Suppose that there are two cascades: misinformation $\mathbb{M}$ and positive cascade $\mathbb{P}$, with seed sets $M \subseteq V$ (attacker) and $P \subseteq V$ (protector), respectively. We speak of each node as being the state of $\mathbb{M}$-active, $\mathbb{P}$-active, or inactive. Following the triggering model [9, 12], the diffusion process unfolds as follows:

- **Initialization**: Each node $u$ samples a subset $A_u \subseteq N_u^-$ from $\mathcal{N}_u$. Each edge $(u, v)$ samples a real number $t_{(u,v)} > 0$ from $\mathcal{T}_{(u,v)}$.

- **Time** 0: The nodes in $M$ (resp, $P$) are $\mathbb{M}$-active (resp,. $\mathbb{P}$-active) at time 0.[1]

- **Time** $t$: When a node $u$ becomes $\mathbb{M}$-active (resp., $\mathbb{P}$-active) at time $t$, each inactive node $v$ such that $u$ in $A_v$ will be activated by $u$ and become $\mathbb{M}$-active (resp., $\mathbb{P}$-active) at time $t + t_{(u,v)}$. Each node will be activated by the first in-neighbor attempting to activate them and never deactivated. When a node $v$ is activated by two or more in-neighbors with different states at the same time, $v$ will become $\mathbb{M}$-active. [2]

**Remark 1.** When there is only one cascade, the above model subsumes classic models, including Discrete-time independent cascade (DIC) model [9], Discrete-time linear threshold (DLT) model [9], Continuous-time independent cascade (CIC) model [12]. An example for illustrating the diffusion process is given in Supplementary A.

## 2.2 Misinformation Prevention and Learning Settings

Given the seed sets $M$ and $P$, we use $f(M, P) : 2^V \times 2^V \to \mathbb{R}$ to denote the expected number of the nodes that are **not** activated by the misinformation and call $f$ the *prevention function*. Formally, they form a class of functions.

**Definition 1** (Class $\mathcal{F}_{\mathrm{PF}}$). Over the choices of $\mathcal{N}_u$ and $\mathcal{T}_{(u,v)}$, we use $\mathcal{F}_{\mathrm{PF}}$ to denote the class of the prevention functions, i.e.,

$$\mathcal{F}_{PF} := \Big\{ f(M, P) : 2^V \times 2^V \to \mathbb{R} \mid \mathcal{N}_u \text{ for each } u; \mathcal{T}_{(u,v)} \text{ for each } (u, v) \Big\}. \tag{1}$$

When the misinformation $M$ is detected, our goal is to launch a positive cascade such that the misinformation can be maximally prevented [7, 13, 14].

**Problem 1** (**Misinformation Prevention**). Under a budget constraint given by $k \in \mathbb{Z}^+$, the misinformation prevention problem aims to compute

$$F(M) := \underset{P \subseteq V \setminus M, \ |P| \leq k}{\arg \max} f(M, P | \emptyset) := f(M, P) - f(M, \emptyset). \tag{2}$$

In this paper, we assume that the social graph $G$ is known but the diffusion model (i.e., $\mathcal{N}_u$ and $\mathcal{T}_{(u,v)}$) is unknown, and given a new attacker $M$, we aim to solve Problem 1 from historical data: a collection of samples $\mathcal{S} = \{(M_i, P_i)\}_{i=1}^n$ where $P_i$ is the optimal or suboptimal solution to Problem 1 associated with input $M_i$. That is, we aim to learn a strategy $F^* : 2^V \to 2^V$ that computes the protector $F^*(M)$ for a future attacker $M \subseteq V$, hoping that $F^*(M)$ can maximize $f(M, P)$ with respective to $P$. Since $f(M, P)$ is unknown to us, $F^*(M)$ is examined by the training pairs. For a training pair $(M, P)$, we consider a function $L(P, S)$ that quantifies the loss for using some $S \subseteq V$ instead of $P$ as the protector. Assuming that the attacker $M$ of the misinformation follows an unknown distribution $\mathcal{M}$, we aim to learn a $F^*$ such that the risk $\int_{2^V} L\big(F(M), F^*(M)\big) d\mathcal{M}(M)$ is minimized, and we attempt to achieve this by minimizing the empirical risk

$$\mathcal{R}_{\mathcal{S}} = \frac{1}{n} \sum_i L(P_i, F^*(M_i)). \tag{3}$$

# 3 StratLearner

The overall idea is to learn a scoring function $f^*$ such that $\arg \max_{P \subseteq V, \ |P| \leq k} f^*(M, P)$ can be a good protector. Note that the prevention function $f$ itself is the perfect score function, but it is not known to us and no data is available for learning it. Nevertheless, we are able to construct a hypothesis space that not only covers the class of prevention function (Sec. 3.1) but also enables simple and robust learning algorithm for searching a scoring function within it (Sec. 3.2).

## 3.1 Parameterization

To construct the desired hypothesis space, let us consider a function class derived through distance functions on subgraphs.

**Definition 2** (Class $\mathcal{F}_\Phi$). Let $\Psi$ be the set of the weighted subgraphs of $G$ over all possible weights and structures, and let $\Phi$ be the set of all distributions over $\Psi$. For each subgraph $g \in \Psi$ and $v \in V$, define that $f_g(M, P | \emptyset) := \sum_{v \in V} f_g^v(M, P | \emptyset)$ with

$$f_g^v(M, P | \emptyset) := \begin{cases} 1 & \mathrm{dis}_g(P, v) < \mathrm{dis}_g(M, v) \text{ and } \mathrm{dis}_g(M, v) \neq \infty \\ 0 & \text{otherwise} \end{cases} \quad \text{(distance function)} \tag{4}$$

where we have $\mathrm{dis}_g(S, v) := \min_{u \in S} \mathrm{dis}_g(S, v)$ and $\mathrm{dis}_g(u, v)$ is the length of the shortest path from $u$ to $v$ in $g$. The class $\mathcal{F}_\Phi$ is defined as $\mathcal{F}_\Phi := \Big\{ \int_\Psi \phi(g) \cdot f_g(M, P | \emptyset) \, dg \mid \phi \in \Phi \Big\}$

**Theorem 1.** $\mathcal{F}_{\mathrm{PF}}$ *is a subclass of* $\mathcal{F}_\Phi$.

The above result indicates that the prevention function can be factorized as an affine combination of the distance functions (i.e. $f_g^v(M, P|\theta)$) over subgraphs with weights given by some $\phi(g)$. While the class $\mathcal{F}_\Phi$ is still not friendly for searching as no parameterization of $\phi(g)$ is given, the function therein can be further approximated by using the subgraphs randomly drawn from some fixed distribution in $\Phi$, as shown in the following.

**Definition 3** (Class $\mathcal{F}_{\mathbf{G}}$). For a subset $\mathbf{G} = \{g_1, ..., g_K\} \subseteq \Psi$, let us consider the function class

$$\mathcal{F}_{\mathbf{G}} := \Big\{ \sum_{i=1}^{K} w_i \cdot f_{g_i}(M, P|\emptyset) \mid w_i \in \mathbb{R} \Big\}. \tag{5}$$

Let $\phi^*$ be any distribution in $\Phi$ with $\phi^*(g) > 0$ for each $g \in \Psi$, and let $\mathbf{G} = \{g_1, ..., g_K\}$ be a collection of random subgraphs generated iid from $\phi^*$. The following result shows the convergence bound for approximating functions in $\mathcal{F}_\Phi$ via functions in $\mathcal{F}_{\mathbf{G}}$, which is inspired by standard analysis of random features [15].

**Theorem 2.** *Let* $\chi$ *be any distribution over* $2^V \times 2^V$ *and* $\epsilon, \delta > 0$ *be the given parameters. For each* $f_1 \in \mathcal{F}_\Phi$ *associated with certain* $\phi_1 \in \Phi$, *when* $K$ *is no less than*

$$\max(2\ln\frac{1}{\delta}, 1) \cdot \frac{C^2 |V|^2}{\epsilon^2}$$

*with probability at least* $1 - \delta$ *over* $g_1, ..., g_K$, *there exists a* $f_2 \in \mathcal{F}_{\mathbf{G}}$ *such that*

$$\sqrt{\int_{2^V \times 2^V} \Big(f_2(x) - f_1(x)\Big)^2 d\chi(x)} \le 2\epsilon, \tag{6}$$

*where* $C := \sup_g \frac{\phi_1(g)}{\phi^*(g)}$ *measures the deviation between* $\phi_1$ *and* $\phi^*$.

Theorems 1 and 2 together imply that each prevention function in $\mathcal{F}_{\mathrm{PF}}$ can be well-approximated by some function in $\mathcal{F}_{\mathbf{G}}$ provided that $\mathbf{G}$ had a sufficient number of random graphs and the weights were correctly chosen. Given that the underlying prevention function is the perfect scoring function, we now have a good reason to search a scoring function in $\mathcal{F}_{\mathbf{G}}$, and we will do so by learning the weights $w_i$, guided by the empirical risk Eq. (3). Now let us assume that the subgraphs $\{g_1, ..., g_K\}$ have been generated, and we focus on learning the weights.

### 3.2 Margin-based Structured Prediction

Given the subgraphs $\mathbf{G} = \{g_1, ..., g_K\}$, according to Eq. (5), our scoring function takes the form of $\mathbf{w}^\mathsf{T} \mathbf{G}(M, P)$ where we have defined $\mathbf{G}(M, P)$ as $\mathbf{G}(M, P) := \Big(f_{g_1}(M, P|\emptyset), ..., f_{g_K}(M, P|\emptyset)\Big)$ and $\mathbf{w} \in \mathbb{R}^K$ are the parameters to learn. For a collection of training pairs $\{(M_i, P_i)\}_{i=1}^n$, the condition of zero training error requires that $\mathbf{w}^\mathsf{T} \mathbf{G}(M, P)$ identifies $P_i$ to be the best protector corresponding to $M_i$, and it is therefore given by the constrains

$$\mathbf{w}^\mathsf{T} \mathbf{G}(M_i, P_i) \ge \mathbf{w}^\mathsf{T} \mathbf{G}(M_i, S), \ \forall i \in [n], \ \forall S : |S| \le k \text{ and } S \ne P_i. \tag{7}$$

In addition, we requires that the weights are non-negative for several reasons. First, the proof of Theorem 2 tells that non-negative weights are sufficient to achieve the convergence bound, so such a requirement would not invalidate the function approximation guarantees. Second, as discussed later in this section, restricting the weights to be non-negative can simplify the inference problem. Finally, as observed in experiments, such a constraint can lead to a fast convergence in the training process, without scarifying the performance. In the case that Eq. (7) is feasible but the solution is not unique, we aim at the solution with the maximum margin. The standard analysis of SVM yields the following quadratic programming:

$$\begin{aligned} \min \quad & \frac{1}{2} \|\mathbf{w}\|_2^2 \\ \text{s.t.} \quad & \mathbf{w}^\mathsf{T} \mathbf{G}(M_i, P_i) - \mathbf{w}^\mathsf{T} \mathbf{G}(M_i, S) \ge 1, \ \forall S : |S| \le k \text{ and } S \ne P_i; \\ & \mathbf{w} \ge 0. \end{aligned}$$

---
**Algorithm 1** Modular-Modular Procedure
---
1: **Input:** $H_{(M,P)}(S)$;
2: $X_0 = P$;
3: **repeat**
4:     $X_{t+1} \leftarrow \arg\min_{|S|=k} \overline{H}^{X_t}_{(M,P)}(S)$;
5:     $t \leftarrow t+1$;
6: **until** stop criteria met;
---

In general, the loss function $L(P, S)$ can be derived from the similarity functions $\mathrm{SIM}(P, S)$ by $L(P, S) := \mathrm{SIM}(P, P) - \mathrm{SIM}(P, S)$, where $\mathrm{SIM}(P, S) \geq 0$ has a unique maximum at $S = P$. For example, the Hamming loss is given by the similarity function $\mathbb{1}(S = P)$. For the MP problem, since the graph structure is given, we can measure the similarity of two sets in terms of the overlap of their neighborhoods. Specifically, for each $S \subseteq V$ and $j \in [n]$, we denote by $H_S^j \subseteq V$ the set of the nodes within $j$ hop(s) from any node in $S$, including $S$ itself, and the similarity between two sets $V_1$ and $V_2$ can be measured by $\mathrm{SIM}_{hop}^j(V_1, V_2) := |H_{V_1}^j \cap H_{V_2}^j|$. We call the loss function derived from such similarities as $j$-*hop loss*.

Incorporating the loss function into the training process by re-scaling the margin [16], we have

$$
\begin{aligned}
\min \quad & \frac{1}{2}\|\mathbf{w}\|_2^2 + \frac{C}{2n}\sum_{i=1}^n \xi_i \\
\text{s.t.} \quad & \mathbf{w}^\mathsf{T}\mathbf{G}(M_i, P_i) - \mathbf{w}^\mathsf{T}\mathbf{G}(M_i, S) \geq \alpha \cdot L(P_i, S) - \xi_i, \ \forall i \in [n], \ \forall S : |S| \leq k, S \neq P_i; \quad (8) \\
& \mathbf{w} \geq 0.
\end{aligned}
$$

where $\alpha$ is a hyperparameter to control the scale of the loss. While this programming consists of an exponential number of constraints for each pair $(M_i, P_i)$, these constraints are equivalent to

$$
\min_{|S| \leq k} \alpha \cdot \mathrm{SIM}(P_i, S) - \mathbf{w}^\mathsf{T}\mathbf{G}(M_i, S) \geq \alpha \cdot \mathrm{SIM}(P_i, P_i) - \mathbf{w}^\mathsf{T}\mathbf{G}(M_i, P_i) - \xi_i.
$$

Therefore, the number of constraints can be reduced to polynomial provided that

$$
\min_{|S| \leq k} H_{(M,P)}(S) := \alpha \cdot \mathrm{SIM}(P, S) - \mathbf{w}^\mathsf{T}\mathbf{G}(M, S) \qquad \text{(loss-augmented inference)} \qquad (9)
$$

can be easily solved, which is the loss-augmented inference (LAI) problem. Unfortunately, such a task is not trivial, even under the Hamming loss.

**Theorem 3.** *The loss-augmented inference problem is NP-hard under the hamming loss or $j$-hop loss. Furthermore, it cannot be approximated within a constant factor under the $j$-hop loss unless $NP$ belongs to $DTIME(n^{\mathrm{poly}\log n})$.*

For the hamming loss, minimizing $H_{(M,P)}(S)$ is simply to maximize $\mathbf{w}^\mathsf{T}\mathbf{G}(M, S)$, which is a submodular function (See proof of Theorem 4), and thus we can utilize the greedy algorithm for an $(1 - 1/e)$-approximation [17]. For the $j$-hop loss, the next result reveals a useful combinatorial property of $H_{(M,P)}(S)$ for solving the LAI problem.

**Theorem 4.** *For each $X \subseteq V$, there exists a polynomial-time computable modular upper bound $\overline{H}^X_{(M,P)}(S)$ of $H_{(M,P)}(S)$ that is tight at $X$.*

This result immediately yields a heuristic algorithm for minimizing $H_{(M,P)}(S)$, as shown in Alg. 1. The algorithm is adapted from the modular-modular procedure for DS programming [18], and it guarantees that $H_{(M,P)}(S)$ is decreased after each iteration.

**Property 1.** Alg. 1 guarantees that $H(X_{t+1}) < H(X_t)$, and each iteration takes $O(K|V|^2 + K|V||E|)$.

Once the LAI problem is solved, the weights $\mathbf{w}$ can be learned using standard structural SVM. We adopt the one-slack cutting plane algorithm [19]. See Alg. 2 in Supplementary C.

### 3.3 StratLearner

Putting the above modules together, we have the following learning strategy: given the social graph and a collection of samples $\{(M_i, P_i)\}_{i=1}^n$, (a) select a distribution $\phi^*$ in $\Phi$ and a loss function; (b) generate $K$ random subgraphs $\{g_1, ..., g_K\}$ using $\phi^*$; (c) run the one-slack cutting plane algorithm to obtain $\mathbf{w} = \{w_1, ..., w_K\}$, where the LAI problem is solved by Alg. 1. Given a new attacker $M$, the protector is computed by $\arg\max_{S \subseteq V, |S| \leq k} \mathbf{w}^\mathsf{T} \mathbf{G}(M, S)$, which is the cardinality-constrained submodular maximization problem and therefore can be approximated again by the greedy algorithm [17]. Here we see that enforcing the weights $\mathbf{w}$ to be nonnegative can make this problem much more tractable, as otherwise, the objective function would not be submodular.

**Remark 2.** Alg. 1 is conceptually simple but practically time-consuming. One can use a limit on the iterations as a simple stop criteria. In our experiment, using only one iteration in each run of Alg. 1 is sufficient to achieve a high training efficacy. For selecting $\phi^*$, the requirement that $\phi^*(g) > 0$ for each $g \in \Psi$ is more technical than practical. Given that no prior information of the diffusion model is available, generating subgraphs uniformly at random is a natural choice, which has been effective in our experiments.

## 4 Experiments

The experiment aims to explore: (a) the performance of StratLearner compared with other possible methods in terms of maximizing $f(M, P|\emptyset)$; (b) the number of features and training pairs needed by StratLearner to achieve a reasonable performance; (c) the impact of the distribution $\phi$ used for generating random subgraphs. The implementations are maintained online [20].

### 4.1 Settings

**Social Graph and Diffusion Model.** We adopt three types of social graphs: a Kronecker graph [21] with 1024 nodes and 2655 edges, an Erdős-Rényi graph with 512 nodes and 6638 edges, and a power-law graph [22] with 768 nodes and 1532 edges. Following the classic triggering model [23, 24], the transmission time $\mathcal{T}_{(u,v)}$ of each edge $(u, v)$ follows a Weibull distribution with parameters randomly selected from $\{1, ..., 10\}$, and for each $u$, we have $\mathcal{N}_u(S) \begin{cases} 1/d_v & S = \{v\}, v \in N_u^- \\ 0 & \text{otherwise} \end{cases}$ with $d_v$ being the in-degree of $v$. For each attacker $M$, the budget $k$ of the protector $P$ is $|M|$.

**StratLearner.** Each subgraph is generated by selecting each edge independently at random with a probability of 0.01, where each selected edge has a weight of 1.0. We denote such a distribution as $\phi_{0.01}^{1.0}$. The number of subgraphs (i.e. features) is enumerated from $\{100, 400, 800, 1600\}$. We adopt the one-hop loss, and the hyperparameter $\alpha$ in Eq. (8) is fixed as 1000.

**Other Methods.** To set some standards, we denote by **Rand** the method that randomly selects the protector. Since the graph structure is known to us, we adopt two popular graph-based methods: HighDegree (**HD**), which selects the nodes with the highest degree as the protector, and Proximity (**Pro**), which selects the neighbors of the attacker as the protector. Recall that our problem can be treated as a supervised learning problem from $2^V$ to $2^V$, so off-the-shelf learning methods are also applicable. In particular, we have implemented Naive Bayes (NB), MLP, Graph Convolutional Network (GCN) [11], and Deep Set Prediction Networks (DSPN) [10]. GCN can make use of graph information, and DSPN is designed to process set inputs.

**Training and Evaluation.** The size of each attacker $M$ is randomly generated following the power-law distribution with parameter 2.5, and the nodes in $M$ are selected uniformly at random from $V$. The best protector $P_{true}$ is computed using the method in [25] which is one of the algorithms for Problem 1 that gives the best possible approximation ratio. In each run, the training and testing set, given their sizes, are randomly selected from a pool of 2500 pairs, where the training size is enumerated in $\{270, 540, 1080, 2160\}$ and the testing size is 270. The subgraphs used in StratLearner are also randomly generated in each run. For each method, the whole training and testing process is repeated five times, and we report the average results with standard deviations. For each predicted protector $P_{pred}$, its quality is measured by the *performance ratio* $\frac{f(M,P_{pred}|\emptyset)}{f(M,P_{true}|\emptyset)} \in [0, 1]$, where $f(M, P|\emptyset)$ is computed using 10000 simulations. Higher is better.

Table 1: **Main Result.** Each cell shows the mean of performance ratio with the standard deviation.

| Dataset | | StratLearner ($\phi_{0.01}^{1.0}$) | | | | ML Methods | | | |
|---------|------|------|------|------|------|------|------|------|------|
| | | 100 | 400 | 800 | 1600 | NB | MLP | GCN | DSPN |
| Kro-necker | 270 | 0.699 (8E-3) | 0.759 (7E-3) | 0.785 (1E-2) | 0.810 (9E-3) | 0.643 (3E-2) | 0.607 (2E-2) | 0.650 (2E-3) | 0.659 (9E-3) |
| | 540 | 0.707 (5E-3) | 0.743 (8E-3) | 0.780 (9E-3) | 0.813 (7E-3) | 0.657 (5E-3) | 0.602 (9E-3) | 0.653 (1E-3) | 0.650 (1E-2) |
| | 1080 | 0.708 (2E-2) | 0.760 (1E-2) | 0.782 (8E-3) | 0.817 (5E-3) | 0.658 (5E-3) | 0.632 (2E-2) | 0.657 (1E-3) | 0.650 (1E-2) |
| | 2160 | 0.701 (1E-2) | 0.756 (1E-2) | 0.792 (5E-3) | 0.821 (8E-3) | 0.655 (3E-3) | 0.661 (1E-2) | 0.648 (1E-3) | 0.666 (6E-3) |
| | | Other Methods: | **Rand**: 0.190 (5E-3) | | **HD**: 0.639 (4E-3) | | **Pro**: 0.670 (6E-3) | | |
| Power-law | 270 | 0.707 (1E-2) | 0.839 (6E-3) | 0.881 (1E-2) | 0.902 (8E-3) | 0.272 (1E-2) | 0.271 (2E-3) | 0.271 (1E-3) | 0.242 (3E-2) |
| | 540 | 0.686 (2E-2) | 0.858 (8E-3) | 0.878 (2E-2) | 0.909 (9E-3) | 0.294 (1E-2) | 0.327 (2E-3) | 0.279 (6E-4) | 0.247 (1E-2) |
| | 1080 | 0.680 (4E-2) | 0.823 (2E-2) | 0.890 (4E-3) | 0.920 (7E-3) | 0.294 (1E-2) | 0.418 (2E-2) | 0.281 (8E-4) | 0.242 (2E-2) |
| | 2160 | 0.682 (1E-2) | 0.853 (2E-2) | 0.889 (1E-2) | 0.911 (3E-3) | 0.302 (3E-3) | 0.489 (1E-2) | 0.275 (6E-4) | 0.235 (1E-2) |
| | | Other Methods: | **Rand**: 0.047 (4E-3) | | **HD**: 0.318 (1E-3) ; | | **Pro**: 0.770 (8E-3) | | |
| Erdős-Rényi | 270 | 0.661 (2E-2) | 0.853 (6E-3) | 0.873 (1E-2) | 0.892 (3E-3) | 0.106 (5E-2) | 0.246 (2E-2) | 0.085 (6E-4) | 0.088 (1E-2) |
| | 540 | 0.673 (2E-2) | 0.861 (1E-2) | 0.876 (6E-3) | 0.897 (1E-2) | 0.104 (5E-3) | 0.340 (2E-2) | 0.088 (1E-3) | 0.095 (7E-3) |
| | 1080 | 0.688 (3E-2) | 0.844 (9E-3) | 0.870 (1E-2) | 0.899 (8E-3) | 0.111 (6E-3) | 0.410 (2E-2) | 0.091 (5E-4) | 0.090 (4E-3) |
| | 2160 | 0.674 (2E-2) | 0.857 (2E-2) | 0.873 (5E-3) | 0.903 (3E-3) | 0.115 (2E-3) | 0.484 (2E-2) | 0.101 (8E-4) | 0.090 (8E-3) |
| | | Other Methods: | **Rand**: 0.052 (2E-2) | | **HD**: 0.102 (5E-3) | | **Pro**: 0.776 (5E-3) | | |

The details of data generation and the implementations of the tested methods can be found in Supplementary E, which also includes the result on a Facebook graph.

## 4.2 Observations

**On StratLearner.** The main results are given in Table 1. We see that StratLearner performs better when more training examples or more features are given, and it is pretty robust in terms of deviation. In addition, StratLearner is more sensitive to the number of features than to the number of training examples - the performance ratio does not increase much when more training examples are given but increases significantly with more features.

**Comparison between Different Methods.** With 400 features from $\phi_{0.01}^{1.0}$, StratLearner has already outperformed all other methods, regardless of the types of the social graph. Plausibly, DSPN and NB are unable to utilize the information of the social graph; GCN is unable to process set structures; HD and Pro ignore the training data. While GCN can make use of the social graph, it merely uses the adjacency between nodes without considering the triggering model. In contrast, StratLearner samples subgraphs and seeks the best combination of them through learning the weights, which, according to Theorem 2, is essentially to approximate the diffusion process under the triggering model. This enables StratLearner to leverage the social graph to learn the unknown parameter in a more explicit way. In another issue, StratLearner, with a moderate number of features, can achieve a performance ratio no less than 0.7 on all the three graphs, but other learning methods (i.e., NB, MLP, GCN) are quite sensitive to the graph structure. In particular, they perform relatively well on Kronecker but poorly on Power-law and Erdős-Rényi. For example, MLP can achieve a ratio comparable to that of HD and Pro on Kronecker, but it is not much better than Rand on Erdős-Rényi. For graph-based methods, HD is also sensitive to the graph structure, while Pro can consistently offer moderate performance, though worse than StratLearner. Overall, the performance of StratLearner is exciting.

**The Impact of $\phi$.** One interesting question is how the distribution used for generating random subgraphs may affect the performance of StratLearner. First, to test the density of the subgraphs, we consider two distributions $\phi_{0.005}^{1.0}$ and $\phi_{0.1}^{1.0}$, where each edge is selected with probability, respectively, 0.005 (less dense) and 0.1 (more dense), with edge weights remaining as 1.0. The results of this part are given in Fig. 2. Comparing $\phi_{0.005}^{1.0}$ and $\phi_{0.1}^{1.0}$ to $\phi_{0.01}^{1.0}$, on Power-law and Erdős-Rényi, we observe an increased performance ratio when the subgraphs become denser, but on Kronecker, decreasing the density also results in a better performance ratio. We can imagine that increasing the subgraph density

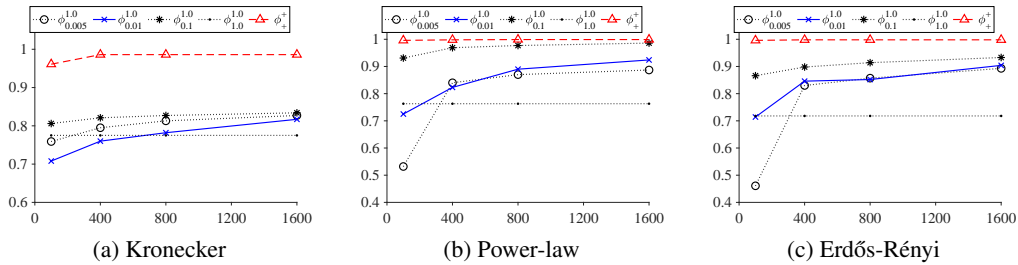

Figure 2: StratLearner with different $\phi$. The y-axis denotes the performance ratio and the x-axis denotes the number of features. Each graph plots five curves for $\phi_{0.005}^{1.0}$, $\phi_{0.01}^{1.0}$, $\phi_{0.1}^{1.0}$, $\phi_{1.0}^{1.0}$ and $\phi_{+}^{+}$, respectively. The precise values are given in Table 2 in Supplementary E.

does not necessarily increase the performance. Considering the extreme setting $\phi_{1.0}^{1.0}$ where each edge is always selected, since there is only one feature, StratLearner reduces to simply maximizing the distance function over the entire graph with uniform weight. As we can see from Fig. 2, StratLearner does not perform well under $\phi_{1.0}^{1.0}$. This is very intuitive as the searching space is too simple to find a good scoring function. Second, we leak some information of the underlying model to $\phi_{0.1}^{1.0}$ and construct $\phi_{+}^{+}$ where the edge is selected with a probability of $1/d_v$, exactly the same as that in the underlying model, with edge weights sampled from their associated Weibull distributions. While $\phi_{+}^{+}$ is not obtainable under our learning setting, the goal here is to verify that StratLearner can benefit more from such cheat subgraphs. Indeed, as shown in Fig. 2, StratLearner can produce the protector that is almost as good as the optimal one. With only 100 features from $\phi_{+}^{+}$, the performance ratio is no less than 0.95 on all three graphs. This confirms that StratLearner does work the way it is supposed to, and it also suggests that such prior knowledge, if any, can be easily handled by StratLearner.

## 5 Further Discussions and Related Work

**Random Features.** Our parameterization method is inspired by the technique of random Fourier features [15, 26], which is an effective method for many learning problems (e.g., [27, 28, 29]). In particular, we show that subgraph sampling can be used to generate random features, and a subtle combination of them can give a kernel function $\mathbf{w}^\top \mathbf{G}(M, P)$ that coincides with the triggering model. This suggests a new way of putting graphs into a learning process, and it is different from other methods like graph neural networks or attentions that often use the entire graph.

**Set Function Learning.** Our problem can be taken as a set function learning problem with sets as input and output. A learning method to solve such problems should respect the set structure invariant to permutation, but neural networks often take vectors as input and their output is sensitive to the positions of the input values. One possible method is to use operations like $sum$ or $max$ that are permutation-invariant [30], and another idea is to enforce the network to learn permutation-invariant representations [10]. Our method is different. StratLearner is invariant to permuting the input set because the constructed kernel function $\mathbf{G}(M, P)$ is combinatorial as it is a set function; it is also invariant to permuting the output set because the inference method is also a combinatorial algorithm that directly outputs a set. In fact, set algorithms are conceptually permutation-invariant operations that generalize $sum$, $avg$ or $max$.

**Misinformation Prevention and Learning Diffusion Models.** Kempe et al. [9] formulate the discrete triggering model, and Du *et al.* [12] later propose the continuous model for modeling information diffusion. The MP problem is first formulated by Budak *et al.* [7]. Even if the diffusion model is given, the MP problem is still challenging because it is NP-hard [7] and its objective function is #P-hard to compute [31]. Later in [32], the authors study the problem of identifying the best intervention period based on the Hawkes process. Learning the diffusion model from real data is another relevant research branch [33, 34, 35, 36]. Du et al. [23] design an algorithm to learn the diffusion function without knowing the type of the diffusion models; He *et al.* [24] study the same problem but assuming the information is incomplete; Kalimeris *et al.* [37] propose a method that parameterizes each edge using the same hyperparameter. Different from the above works, this paper aims to learn a solution to the MP problem, and it does not attempt to learn the diffusion model.

## Broader Impact

The work in this paper focuses on operational diffusion models without specifying a particular social network platform. Our work proposes a framework for computing protectors, but more importantly and broadly, it suggests a new method for solving learning problems by integrating graph input into the structured prediction. In addition, we do not anticipate any bias in the data used for experiments because the involved subgraphs, underlying triggering model, training examples, and training-testing partition were all randomly determined with enough repetitions. One exception is that we have considered only three graph types, Kronecker, Power-law, and Erdős-Rényi, which may lead to the bias on the graph structure. However, given that the results of StratLearner are robust over these graphs, we believe the observations can be generalized to other graph structures.

## Acknowledgments and Disclosure of Funding

This work is supported by a start-up package from the University of Delaware.

## Footnotes

[1]Without loss generality, we assume that $M \cap P = \emptyset$.

[2]This setting is not critical. See Supplementary D for a discussion.

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
