[Supplementary Material · neurips_2020.pdf]

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

# StratLearner: Learning a Strategy for Misinformation Prevention in Social Networks
# (Supplementary Material)

## A    Diffusion Process

The first graph in Fig. 3 shows a triggering model where each node is associated with a distribution over its in-neighbors and each edge holds a distribution showing the activation time. The second graph shows one possible scenario after initialization, a weighted subgraph where $(u, v)$ is in the graph iff $u \in A_v$. Based on the initialization in the second graph, the third graph shows the diffusion results under $M = \{a\}$ and $P = \{b\}$; the fourth graph shows the case when $M = \{a\}$ and $P = \{f\}$.

Figure 3: Diffusion Process.

## B    Proofs

### B.1    Proof of Theorem 1

Note that the initialization step in the diffusion process is equivalent to generating a weighted subgraph with edges $\cup_{u \in V}\{(v, u)|v \in A_u\}$ in which each edge $e$ has a weight of $t_e$. Therefore, each diffusion model defines a distribution $\phi$ over $\Psi$, and it suffices to prove $f(M, P|\emptyset) = \int_g \sum_{v \in V} \phi(g) \cdot f_g^v(M, P|\emptyset) dg$. Exchanging the summation with integration and using the linearity of expectation, it suffices to prove that $\int_\psi \phi(g) \cdot f_g^v(M, P|\emptyset) dg$ is equal to the probability that $v$ will not be $\mathbb{M}$-active with $P$ but would have been $\mathcal{M}$-active without $P$. Since the rest of the diffusion is determined after realization, it is left to prove that for each $v^* \in V$, $f_g^{v^*}(M, P|\emptyset) = 1$ iff, under the initialization corresponding to $g$, (a) $v^*$ will not be $\mathbb{M}$-active with $P$ and (b) $v^*$ will be $\mathbb{M}$-active without $P$. This can be easily established from the facts: (a) a node $v$ can be activated by one cascade only if there is a path from the seed nodes to $v$; (b) the node will be activated by the first cascade arriving them; (c) the arrival time each of cascade depends on the length of the shortest path from the source node to $v$. A formal argument can be obtained using a reduction from the $\arg\min_{u \in M \cup P} \mathrm{dis}(u, v)$ to $v$ along the shortest path.

### B.2    Proof of Theorem 2

The proof uses the McDiarmid's Inequality.

**Definition 4** (McDiarmid's Inequality [38]). Let $X_1,..., X_m$ be independent random variables with domain $\mathcal{X}$. Let $f : \mathcal{X}^m \to \mathbb{R}$ be a function that satisfies $|f(x_1, ..., x_i, ..., x_m) - f(x_1, ..., x_i', ..., x_m)| \leq c_i$ for each $i$ and $x_1, ..., x_m, x_i' \in \mathcal{X}$. The for each $\epsilon > 0$, we have $\Pr[f - \mathbb{E}[f] \geq \epsilon] \leq \exp\left(\frac{-2\epsilon^2}{\sum c_i^2}\right)$.

Consider the function $h_i = \frac{\phi_1(g_i)}{\phi^*(g_i)} f_{g_i}(M, P)$ for $i \in [K]$, and the average function

$$f_2 = \sum_{i=1}^K \frac{\phi_1(g_i)}{K \cdot \phi^*(g_i)} f_{g_i}(M, P).$$

Note that $f_2 \in \mathcal{F}_\mathbf{G}$ and $\mathbb{E}[h_i(M, P)] = f_1(M, P)$ for each $(M, P)$. Let us denote the interested quantity as

$$\Delta_K(g_1, ..., g_K) = \sqrt{\int_{2^V \times 2^V} \left(f_2(x) - f_1(x)\right)^2 d\chi(x)} = \|f_2 - f_1\|$$

where the norm is taken under the Lebesgue measure associated to measure $\chi$ (the distribution over the pairs $(M, P)$). Now let us consider $\mathbb{E}[\Delta_K(g_1, ..., g_K)]$. The upper bound of $\mathbb{E}[\Delta_K(g_1, ..., g_K)]$ is found by

$$
\begin{aligned}
\mathbb{E}[\Delta_K(g_1, ..., g_K)] &\leq \sqrt{\mathbb{E}\Big[\int_{2^V \times 2^V} \big(f_2(x) - f_1(x)\big)^2 d\chi(x)\Big]} \\
&= \sqrt{\int_{2^V \times 2^V} \mathbb{E}\Big[\big(f_2(x) - f_1(x)\big)^2\Big] d\chi(x)} \\
&= \sqrt{\int_{2^V \times 2^V} \mathbb{E}\Big[(f_2(x))^2\Big] - \mathbb{E}\Big[f_1(x)\Big]^2 d\chi(x)} \\
&\leq \sqrt{\int_{2^V \times 2^V} \frac{1}{K^2} \sum_i \mathbb{E}\Big[\Big(\frac{\phi_1(g_i)}{\phi^*(g_i)} f_{g_i}(x)\Big)^2\Big] d\chi(x)} \leq \frac{C \cdot |V|}{\sqrt{K}}
\end{aligned}
$$

To show the stability of $\Delta_K(g_1, ..., g_K)$, for each $g_1, ..., g_K, g_* \in \Psi$ and $i \in [K]$, replacing $g_i$ by $g_*$, the change is bounded by

$$
\begin{aligned}
&|\Delta_K(g_1, ..., g_i, ...g_K) - \Delta_K(g_1, ..., g_*, ...g_K)| \\
&\{\text{by the reverse triangle inequality}\} \\
&\leq \|f_2(g_1, ..., g_i, ...g_K) - f_2(g_1, ..., g_*, ...g_K)\| \\
&= \sqrt{\int_{2^V \times 2^V} \Big(\frac{\phi_1(g_i)}{K \cdot \phi^*(g_i)} f_{g_i}(M, P) - \frac{\phi_1(g_*)}{K \cdot \phi^*(g_*)} f_{g_*}(M, P)\Big)^2 d\chi(x)} \qquad \leq \frac{2 \cdot C \cdot |V|}{K}
\end{aligned}
$$

By Eq. (6), we have

$$
\begin{aligned}
\Pr[\Delta_K(g_1, ..., g_K) - \epsilon \geq \epsilon] &\leq \Pr[\Delta_K(g_1, ..., g_K) - \frac{C \cdot |V|}{\sqrt{K}} \geq \epsilon] \\
&\leq \Pr[\Delta_K(g_1, ..., g_K) - \mathbb{E}[\Delta_K(g_1, ..., g_K)] \geq \epsilon] \\
&\{\text{McDiarmid's inequality}\} \\
&\leq \exp(\frac{-2K\epsilon^2}{(2 \cdot C \cdot |V|)^2}) \leq \delta.
\end{aligned}
$$

## B.3 Theorem 3

An instance of this problem is given by the social graph $G = (V, E)$, a collection $\{g_1, ..., g_K\}$ of subgraphs, a weight vector $\mathbf{w}$, the budget $k$, and two node sets $M$ and $P$. Recall that the objective function is $\alpha \cdot H_{(M,P)}(S) := \text{SIM}(P, S) - \mathbf{w}^\mathsf{T} \mathbf{G}(M, S)$.

The NP-hardness can be easily established through a reduction from the max k-coverage problem. The max k-coverage problem is given by an element set $P = \{p_1, ..., p_N\}$ and a collection $Q = \{q_1, ..., q_M\} \subseteq 2^P$, and it asks for $l$ sets in $Q$ with the largest union. Setting $K = 1, w_1 = 1$, let us consider the $g_1$ given in Figure 4 where one node is created for each $p_i$ and $q_j$ with an extra node $z$ added to the graph. There is an edge from node $q_j$ to node $p_i$ if and only if element $p_i$ is in set $q_j$, with a weight of 0.5; there is an edge from $z$ to each $p_i$ with a weight of 1. The social graph $G$ can be any supergraph of $g_1$ and $P$ can be any node set that does not contain any node in $g_1$. Setting $k = l$ and $M = \{z\}$, we see that the $S \subseteq Q$ that can minimize $H_{(M,P)}(S)$ corresponds to the one that has the largest union in the max k-coverage problem.

Figure 4: Reduction for NP-hardness.

To prove the approximation hardness, we seek a reduction from the positive-negative set cover (k±PSC) problem.

**Problem 2** (k±PSC problem). An instance of k±PSC is a triplet $(X, Y, \Phi)$ with an integer $l \in \mathbb{Z}^+$, where $X$ and $Y$ are two sets of elements with $X \cap Y = \emptyset$, and $\Phi = \{\phi_1, ..., \phi_q\} \subseteq 2^{X \cup Y}$ is a collection of $q \in \mathbb{Z}^+$ subsets over $X \cup Y$. For each $\Phi^* \subseteq \Phi$, its cost is defined as

$$\text{cost}(\Phi^*) = |X \setminus (\cup_{\phi \in \Phi^*} \phi)| + |Y \cap (\cup_{\phi \in \Phi^*} \phi)|.$$

The k±PSC problem seeks for a $\Phi^* \subseteq \Phi$ with $|\Phi^*| = l$ such that the cost is minimized.

The following hardness of k±PSC follows fairly directly from Miettinen [39].

**Lemma 1** ([39]). *Unless $NP \subseteq DTIME(n^{\text{poly} \log n})$, there exists no polynomial-time approximation algorithm for the k±PSC problem with a ratio of $\Omega(2^{\log^{1-\epsilon} q})$ for each $\epsilon > 0$.*

Given an instance $(X, Y, \Phi)$ of k±PSC, we construct an instance of LAI, as follows. The social graph is show in Fig. 5a composed of the following parts:

- Nodes: There are four groups of nodes $X, X^*, Y$ and $\Phi$, where each node in $X, Y$ and $\Phi$ corresponds to their counterpart in the k±PSC instance, and $X^*$ is a copy of $X$. In addition, there are two extra nodes $z_1$ and $z_2$;

- Edges: edges can be grouped into several parts:
    - There is an edge from $z_2$ to each node in $X^* \cup Y$, and an edge from $z_1$ to each node in $X$;
    - There is an edge from each node in $\Phi$ to each node in $X^*$;
    - There is an edge from each node in $X$ to each node in $X^* \cup Y$.
    - There is an edge $(u, v)$ for each pair of the nodes in $X^* \cup Y$. This part is not shown in the graph.
    - There is an edge from $\phi_i$ to $y_j$ if and only if $y_j \in \phi_i$.
    - There is an edge from $\phi_i$ to $x_j$ if and only if $x_j \in \phi_i$.

(a) Social Graph $G$          (b) The subgraph $g_1$

Figure 5: Reduction for Approximation Hardness.

We again set $K = 1$, $w_1 = 1$, $\alpha = 1$, and $g_1$ is a subgraph of $G$ which includes nodes $\Phi, X$ and $z_1$ and the edges between them, as shown in Fig. 5b. In $g_1$, each edge between $\Phi$ and $X$ has a weight of 0.5; each edge between $X$ and $z_1$ has a weight of 1.0. We set $P$ as $\{z_2\}$ and $M$ as $\{z_1\}$. We consider the one-hop loss and set $l = k$. Due to the edge within $Y$ and those from $Y$ to $X$, to minimize $\text{SIM}(P, S)$ (i.e., the overlap of one-hop neighbors), only the nodes in $\Phi$ should be selected. To maximize $\mathbf{w}^\top \mathbf{G}(M, S) = g_1(M, S)$, according to the construction of $g_1$, we see again that the optimal solution must the nodes in $\Phi$. Therefore, for the LAI problem, it suffices to restrict the node selection in $\Phi$. For each subset $\Phi^* \subseteq \Phi$, $\text{SIM}(P, S)$ is exactly $|X^*|$ plus the number of the nodes in $Y$ that are connected from some node in $\Phi^*$, and $\mathbf{w}^\top \mathbf{G}(M, S)$ is the number of the nodes in $X$ that are connected from some node in $\Phi^*$. Therefore, we have

$$H_{(M,P)}(\Phi^*) = \text{SIM}(P, \Phi^*) - \mathbf{w}^\top \mathbf{G}(M, \Phi^*)$$
$$= |X^*| + |Y \cap (\cup_{\phi \in \Phi^*} \phi)| - |X \cap (\cup_{\phi \in \Phi^*} \phi)|$$
$$= |Y \cap (\cup_{\phi \in \Phi^*} \phi)| + |X \setminus (\cup_{\phi \in \Phi^*} \phi)| = \text{cost}(\Phi^*),$$

which completes the reduction. The reduction yields the hardness result immediately.

## B.4   Proof of Theorem 4

A set function $h$ over a ground set $U$ is submodular if it has a diminishing marginal return, i.e., $h(A+v)-h(A) \leq h(B+v)-h(B)$ for each $B \subseteq A$ and $v \notin A$. $\mathrm{SIM}_{hop}^j$ is submodular as it is a coverage function [40]. We can easily verified that $f_v^g(M, S|\emptyset)$ is also submodular for each subgraph $g$ and $v$, and therefore, $\mathbf{w}^\mathsf{T}\mathbf{G}(M, \Phi^*)$ is submodular as well since it is a sum of submodular functions. It is well-known that submodular functions have both tight modular upper bound and tight modular lower bound. A modular upper bound of $\mathrm{SIM}_{hop}^j$ together with a modular lower bound of $\mathbf{w}^\mathsf{T}\mathbf{G}(M, \Phi^*)$ would give a modular upper bound of $H_{(M,P)}(S)$. In particular, for a general submodular function $h$ over $U$, the constructions can be found in [18], as summarized below.

**Modular Lower Bound [41].**  For a permutation $\sigma(i)$ over $U = [n]$, let us define that $S_\sigma^i := \{\sigma(1), ..., \sigma(i)\}$ and define a mapping $U \to \mathbb{R}$:

$$\Delta_\sigma(\sigma(i)) := \begin{cases} h(S_\sigma^1) & i = 1 \\ h(S_\sigma^i) - h(S_\sigma^{i-1}) & \text{otherwise} \end{cases}.$$

Since $h$ is submodular, for each $X$ and a permutation $\sigma_X$ such that $X = S_\sigma^{|X|}$, the modular function

$$\underline{h}^{\sigma_X}(S) = \sum_{v \in S} \Delta_\sigma(v) \tag{10}$$

satisfies $\underline{h}^{\sigma_X}(S) \leq h(S)$ for each $S \subseteq V$, and $\underline{h}^{\sigma_X}(X) = h(X)$.

**Modular Upper Bound [17, 18].**   For each $X \subseteq U$, a modular upper bound of $h(S)$ is found by

$$\overline{h}^X(S) := h(X) - \sum_{v \in X \setminus S} \Big( h(X) - h(X \setminus \{v\}) \Big) + \sum_{v \in S \setminus X} \Big( h(X \cap S) - h(X \cap S \setminus \{v\}) \Big),$$

satisfying $\overline{h}^X(X) = h(X)$ and $\overline{h}^X(S) \geq h(S)$ for each $S \subseteq V$.

## B.5   Proof of Property 1

The first part follows from the fact that $H(X_{t+1}) \geq H'_{X_t}(X_{t+1}) \geq H'_{X_t}(X_t) = H(X_t)$. Evaluating the function takes $K(|E| + |V|)$ and thus each iteration takes $K|V|(|E| + |V|)$.

# C   Cutting Plane Algorithm

We adopt the one-slack cutting plane algorithm (Alg. 2) in [19] for training our structural SVM, with the only modification that a nonnegative constraint on $\mathbf{w}$ is added.

---

**Algorithm 2** One-slack Cutting Plane

---

1: **Input:** $(M_1, P_1), ..., (M_n, P_n), C, \epsilon, \alpha$;
2: $\mathcal{W} \leftarrow \emptyset$;
3: **repeat**
4:     Solve the QP over constraints $\mathcal{W}$:

$$(\mathbf{w}, \xi) \leftarrow \arg\min \quad \frac{1}{2}\|\mathbf{w}\|_2^2 + C \cdot \xi_i$$
$$\text{s.t.} \qquad \forall(\overline{P}_1, ...\overline{P}_n) \in \mathcal{W}:$$
$$\frac{1}{n}\mathbf{w}^\mathsf{T}\sum_{i=1}^n \big(\mathbf{G}(M_i, P_i) - \mathbf{G}(M_i, \overline{P}_i)\big) \geq \frac{\alpha}{n} \cdot \sum_{i=1}^n L(P_i, \overline{P}_i) - \xi;$$
$$\mathbf{w} \geq 0.$$

5:     **for** $i = 1, ..., n$ **do**
6:         $\hat{P}_i \leftarrow \arg\min_{|S| \leq k} \mathrm{SIM}(P_i, S) - \mathbf{w}^\mathsf{T}\mathbf{G}(M_i, S)$;
7:     $\mathcal{W} \leftarrow \mathcal{W} \cup \{(\hat{P}_1, ...\hat{P}_n)\}$;
8: **until** $\frac{\alpha}{n} \cdot \sum_{i=1}^n L(P_i, \hat{P}_i) - \frac{1}{n}\mathbf{w}^\mathsf{T}\sum_{i=1}^n \big(\mathbf{G}(M_i, P_i) - \mathbf{G}(M_i, \hat{P}_i)\big) \leq \xi + \epsilon$

---

## D  Tie Breaking

The tie breaking in Sec. 2.1 is done by always giving the misinformation a higher priority. If we would do the reverse, the only modification to StratLearner is to replace Eq. (4) with

$$f_g^v(M, P | \emptyset) := \begin{cases} 1 & \mathbf{dis_g(v, P)} \leq \mathbf{dis_g(v, M)} \text{ and } \mathrm{dis}_g(v, M) \neq \infty \\ 0 & \text{otherwise} \end{cases}.$$

## E  Experiments

The used data and source code are available in the supplementary files.

### E.1  Data Generation

The Kronecker graph is generated using SNAP[3] with parameters $[0.9, 0.6; 0.6, 0.1]$. The power-law graph and the Erdős-Rényi graph are generated using NetworkX [42]. Each edge follows the Weibull distribution $\frac{\beta}{\alpha}(\frac{t}{\alpha})^{\beta-1}\exp(-(\frac{t}{\alpha})^{\beta})$ where $\alpha$ and $\beta$ are selected from $\{1, ..., 10\}$ uniformly at random.

To generate one pair of attacker and protector, we first sample the size of the attacker from the power-law distribution with a parameter 2.5. Given the size of the attacker $M$, the nodes in $M$ are randomly selected from $V$. Given the attacker $M$, the protector $P$ is computed using the method in [25]. Repeating this process, we generate a pool of 2500 pairs for each graph.

### E.2  Method Implementations

**StratLearner.** The one-slack cutting plane algorithm is implemented based on Pystruct [43] with hyperparameters $\epsilon = 0.001$ and $C = 0.01$.

**MLP and GCN.** For MLP, we adopt three hidden layers of size $(512, 512, 256)$ with ReLU as the activation function. The node sets are encoded as one-hot vectors, and the loss function is the pointwise cross-entropy between the output layer and the truth vector, plus the L2 regularizer. We use Adam optimizer with drop rate 0.5, and the learning rate is 0.001 with exponential decay. We adopt the valina GCN model [11] with two GCN layers followed by our MLP. Since the model in [11] was for semi-supervised learning, we slightly modify the flow to make it work for supervised learning. Other settings are the same as those in MLP.

**DSPN.** Dspn is proposed in [30] where the main modules are input encoder, set encoder, and set decoder. Given an attacker, we encode it as a set of elements where the feature of each element is the associated one-hot vector. The input encoder and set encoder are MLP with three hidden layers of size 512. The inner optimization is performed 10 steps with rate $1,000,000$ in each round, and the outer loop is optimized with Adam with a learning rate of 0.01.

### E.3  Detailed Results of Fig. 2.

The precise results in Fig. 2 are given in Table 2.

Table 2: **Testing** $\phi$. 1080 training pairs are used in each experiment.

| $\phi$ | Kronecker | | | | Power-law | | | | Erdős-Rényi | | | |
|---|---|---|---|---|---|---|---|---|---|---|---|---|
| | 100 | 400 | 800 | 1600 | 100 | 400 | 800 | 1600 | 100 | 400 | 800 | 1600 |
| $\phi_{0.005}^{1.0}$ | 0.759 | 0.795 | 0.813 | 0.827 | 0.532 | 0.840 | 0.870 | 0.887 | 0.461 | 0.830 | 0.857 | 0.893 |
| $\phi_{0.01}^{1.0}$ (base) | 0.708 | 0.760 | 0.782 | 0.817 | 0.725 | 0.823 | 0.890 | 0.924 | 0.714 | 0.846 | 0.852 | 0.904 |
| $\phi_{0.1}^{1.0}$ | 0.806 | 0.821 | 0.827 | 0.834 | 0.931 | 0.969 | 0.977 | 0.986 | 0.866 | 0.898 | 0.914 | 0.933 |
| $\phi_{1.0}^{1.0}$ | | 0.775 | | | | 0.763 | | | | 0.748 | | |
| $\phi_+^+$ | 0.961 | 0.986 | 0.986 | 0.986 | 0.996 | 0.998 | 0.999 | 0.999 | 0.996 | 0.998 | 0.998 | 0.998 |

Table 3: **Result on Facebook.**

| StratL | NB | MLP | GCN | DSPN | HD | Pro | Rand |
|---|---|---|---|---|---|---|---|
| 0.725 (1E-2) | 0.662 (6E-3) | 0.651 (5E-3) | 0.625 (2E-3) | 0.446 (2E-3) | 0.656 (8E-3) | 0.170 (1E-2) | 0.011 (8E-3) |

## E.4 Experimental Results on Facebook.

We also tested a Facebook graph with $4,039$ nodes from SNAP[4], where StratLearner is trained with 100 subgraphs from distribution $\phi_{0.1}^{1.0}$ and 270 training examples are used in each learning-based method. Other settings are the same as the experiments in the main paper. The results are given in Table 3. Overall, similar to Table 1 in the main paper, we have the observation that StratLearner outperforms other competitors by an evident margin.