[Reviews · NeurIPS 2020]

Review 1

Summary and Contributions: This paper studies the problem of misinformation prevention in social networks in a setting where the underlying diffusion process of the network is unknown and where the input is previous solutions over that same network. The problem of misinformation prevention in social networks is a variant of the influence maximization problem where an attacker selects a set M of seeds to spread some misinformation and the goal is to select a set P of seeds to spread some protection against the misinformation. Previous work studies this problem assuming that the diffusion process is known. The authors argue that this is unrealistic and consider a setting where the parameters of the diffusion process are unknown and instead only historical data corresponding to previous solutions Pi for some sets Mi is known. The suggested method for this problem, called StratLearner, is to learn a scoring function whose optimal solution is a good protector. The hypothesis space is an interesting family of functions which are affine combinations of distance functions over sampled subgraphs. This family of functions is large enough so that it contains the original prevention functions but also has a nice structure that enables efficient learning. Experiments demonstrate that this approach finds solutions that outperform the solutions found by alternative methods. ***After rebuttal*** I appreciate the clarification about the samples not needing to be optimal but only approximately optimal, this seems realistic.

Strengths: The techniques behind the suggested approach are novel and interesting. In particular, the parameterization for the learning process which uses random features generated by subsampling the network is a nice new idea. I think that considering settings where the parameters of the diffusion process are unknown is a great direction. Strong experimental results that show that this method outperforms a wide range of alternate approaches.

Weaknesses: I agree with the authors that it is unrealistic to know the parameters of the diffusion process, but it is not clear to me that knowing past optimal solutions is realistic, especially since the problem is NP-hard. The introduction could use some additional motivation for the setting considered in this paper. There is a step in the method that I find confusing. Problem (8) introduces some loss functions L(P,S) and it is then argued that the number of constrained can be reduced to be polynomial. Why do these loss functions need to be added? Is it possible to reduce the number of constraints from the previous quadratic programming problem without the L(P, S) terms and solve that problem? It makes sense that with a sufficiently large K, F_PF can be well approximated by some function in F_G. The interesting part is then how large K needs to be. The bound on K contains a C^2 term, it is unclear to me whether C is reasonably large or not.

Correctness: Yes

Clarity: Yes

Relation to Prior Work: Overall, yes, the further discussions and related work sections clearly compares this work to previous work. The only minor comment is regarding the misinformation prevention problem formation as Problem (1), is this the same formulations as in previous work? If yes, the previous work introducing this problem should be cited there.

Reproducibility: Yes

Additional Feedback: It would be nice if the experiments not only contained synthetic graphs but also real world social graphs. The performance ratio is computed by comparing the solution obtained to P_true from [22], is P_true guaranteed to be optimal? Line 99, f*(M) should be f*(M, P)


Review 2

Summary and Contributions: In this paper, the authors proposed a large margin based method to solve the MP (misinformation prevention) problem via learning a scoring function that uses some random features constructed through distance functions. Experiments are conducted on three synthetic datasets and show a reasonable performance against other methods.

Strengths: This paper is well-structured and explains the problem clearly. This paper clearly explains the misinformation prevention problem and its challenges. All the property and theorem are well-defined and put in the logical order. This paper provides a novel way to solve the MP problem. It converts the MP problem into learning a scoring function and solve it with a large margin based method. This paper is sound in technical quality. The setups of the empirical studies are reasonable. The experiment results show StratLearner achieves a reasonable performance compared to the other machine learning-based method and discuss the impact of different distributions of StratLearner.

Weaknesses: All experiments are conducted on synthetic data. The reviewer prefer to see at least one real data experiment. The main concern of the reviewer is the motivation that sets the form of scoring function as the affine combination of distance functions is unclear. The reason to choose such hypothesis space need further justification.

Correctness: The methodology is derived correctly and explained clearly. The experimental setups are reasonable.

Clarity: Yes, the paper is clear and well written.

Relation to Prior Work: The difference between the proposed method and previous work are clear and well-explained.

Reproducibility: Yes

Additional Feedback: A well-structured paper addressing a timely issue of preventing misinformation spread in the social network. The idea of using random subgraphs to construct feature is central to the paper, however, the paper does not offer a reasonable justification with this ideal. All experiments are conducted on three synthetic data. it would perhaps have been worthwhile to experiment with and suggest real-world data. Another observation is that all three datasets are quite small. It seems StratLearner will take some time to get the final protector. Since the real social network is large, not sure if the StratLearner can scale to real-world data. It is better to add experiments to discuss the performance and time concerning the size of a graph.


Review 3

Summary and Contributions: In this paper, the authors study the problem of directly learning strategies for misinformation prevention problem in social network. The author utilized the idea of random graph features to first a linear combination weights for the random graph features as the influence function. Then a submodular optimization is carried out to produce the prevention seed set. The authors carry out extensive experiment on several synthetic cascades with comparison to several baselines. I have read the authors's feedback and will remain my original evaluation.

Strengths: 1. It is an interesting problem to solve diffusion problem under model-free setting. As model misspecification in many cases is inevitable for diffusion modeling. 2. The proposed method has theoretical justification for the function space from random graph features. The paper is technically sound. 3. The authors carry out extensive experiment on synthetic cascades with comparison to neural network-based methods.

Weaknesses: 1. There is no end-to-end approximation guarantee for the proposed method. There is still a large gap between the results to the required guarantee. Theorem 2 only guarantees the existence of function in the space with random features. Several components are missing. (1) The learning algorithm can find the right weight even in the correct function space; (2) One can find the correct proposal distribution \phi. C could easily have exponential dependence on the size of the graph size. (3) The training data generation involves NP-hard problem so the training data is at best approximate. How it effects the learned solution? 2. The problem setting is a little bit artificial. Usually the observation comes from the seeds of both positive and negative information and the outcome of the diffusion. Also, under this case, it would be interesting to compare the proposed method to first learning the influence function under the competitive model from normal observation data and then carry out optimization for prevention. 3. The description for the comparison to NN based method is little bit vague. How is the problem formulated? Do the authors directly train a network with |V| output nodes and use standard classification loss? Also, it would be better if the authors could provide run time experiments. In addition, experiment under different diffusion models would be interesting to see as well.

Correctness: The paper is technically sound with correct empirical evaluation.

Clarity: Overall the paper is well written and easy to follow.

Relation to Prior Work: Yes, the paper provides a satisfying review of related work.

Reproducibility: Yes

Additional Feedback: It should be made clear in Equation (2) that one can not select existing positive seed for prevention based on the model in section 2.1.


Review 4

Summary and Contributions: In this paper, the authors have introduced a method for misinformation prevention on a social graph, through the process of launching a counter cascade. The proposed work seems to be an interesting take on the influence minimization problem in that it casts the problem in the light of learning a solution to the optimization problem using attacker-protector pairs, with no knowledge of the underlying prevention or diffusion function.

Strengths: 1. The proposed method is a new and interesting way to tackle the influence limitation problem, especially one where the diffusion model is unknown, using concepts well-grounded in theory 2. Most questions that arose while reading the main manuscript have been addressed in the supplementary materials. 3. Detailed proofs have been provided by the authors for the claims made. 4. The proposed method demonstrates learning with permutation invariant data, i.e. sets and given that learning with structured data or sets is inherently challenging, this work serves as a good example.

Weaknesses: 1. Although the proposed method demonstrates the use of several recent as well as long-standing methods and results, I do not see why this method of predicting protectors would be chosen given the existence of the hybrid sampling method by Tong & Du [1]. 2. What role do the subgraphs and the subgraph space play? A justification for why subgraphs are considered needs to be given. Are the sets M and P part of the randomly generated subgraph? The use of subgraphs as features in the training process needs to be justified. Also, I would like to see a discussion on the authors’ thoughts regarding why the training process is so much sensitive to the number of such subgraphs. I see the idea of using subgraphs as features has been briefly mentioned in Section 5 but I would like to see an explanation somewhere before (earlier on) in the manuscript, where the idea of using subgraphs was first mentioned. 3. The proposed method seems to have a series of computation heavy steps. How does the performance of StratLearner compare to the other methods mentioned, especially the one by Tong & Du [1]? 4. Experiments on some real datasets would have been interesting. [1] Tong, Guangmo Amo, and Ding-Zhu Du. "Beyond uniform reverse sampling: A hybrid sampling technique for misinformation prevention." IEEE INFOCOM 2019-IEEE Conference on Computer Communications. IEEE, 2019. A few minor concerns: 1. Line 53: “NP-hardness” of what? 2. Line 70: “each inactive node in A_u will be activated by u and become M-active (resp., P-active) at time t+t_(u,v)” - If A_u is a subset of the in-neighbors of u, then how can A_u be activated AFTER u is activated? Should be out-neighbors instead? ************************************************** After reading the rebuttal, I believe authors have responded to some of my questions. However, I agree with some of the concerns brought up by other reviewers. Overall, I would like to keep my score unchanged.

Correctness: Yes, but demonstration using real-world social graphs would have made the claims more convincing.

Clarity: Could be much better. For example: the use of random subgraphs as features has not been justified until much later in the manuscript, and even then, the justification is not convincing enough.

Relation to Prior Work: Yes but I am not convinced that this method of finding good protectors should be used over the method in [1].

Reproducibility: Yes

Additional Feedback:

[Author Response · NeurIPS 2020]

**StratLearner: Learning a Strategy for Misinformation Prevention in Social Networks (Author Response)**

We thank all the reviewers for their time and constructive comments. This letter will first discuss common issues raised
by the reviewers and then respond to individual comments.

**Experiments on real-world graphs (Reviewers 1-4).** We agree that real-
world graphs are worth leveraging for experimentation, and we present
one result (Table A) on a Facebook graph with $4,039$ nodes from SNAP[1],
where StratLearner is trained with $100$ subgraphs from distribution $\phi_{0.1}^{1.0}$
and $270$ training examples are used in each learning-based method. Other
settings are the same as the experiments in the paper. Overall, similar to
Table 1 in the paper, we have the observation that StratLearner outperforms
other competitors by an evident margin. We will include this part in our
paper to strengthen the experimental studies.

**Table A: Results on Facebook**

| StratL | NB | MLP | GCN |
|---|---|---|---|
| 0.725 (1E-2) | 0.662 (6E-3) | 0.651 (5E-3) | 0.625 (2E-3) |

| DSPN | HD | Pro | Rand |
|---|---|---|---|
| 0.446 (2E-3) | 0.656 (8E-3) | 0.170 (1E-2) | 0.011 (8E-3) |

**Motivation of the ideas (Reviewers 2 and 4).** The form of our scoring function is motivated by the fact (Theorem 1)
that the prevention function (which is the perfect scoring function but unknown) can be factorized as an integration of
the distance functions over subgraphs. Because the true distribution is unknown, we attempt to use an affine combination
of random subgraphs to approximate the true distribution, where the weights are adjusted using data; the feasibility of
doing so is partially justified by Theorem 2 stating that the function approximation can be theoretically bounded. We
agree that it would better to highlight the motivations earlier in the manuscript, and we will make revisions accordingly.

**Reviewer 1. a)** It is true that knowing the past optimal solution is not realistic, especially given that the considered
problem is NP-hard, and we wish to note that our framework does not require the samples to be optimal to produce good
protectors. As long as the sample solutions are of high quality, they are sufficient to guide the model to discriminate
between high- and low-quality solutions, which is evidenced by our experiments where the sample solutions are
approximations. **b)** The loss function $L(P, S)$ is introduced to measure the difference between $P$ (ground-truth) and $S$
(prediction). The constraints cannot be (easily) reduced to polynomial even if $L(P, S)$ is constant, because the inference
problem would still be NP-hard (Theorem 3). **c)** We are not able to answer – in theory – how large $K$ needs to be to
ensure the final performance; experimentally, $K = 400$ is sufficient for achieving a decent performance ratio on the
considered datasets. In practice, the most cost-effective $K$ might be determined through cross-validation.

**Reviewer 2.** We do not report the running time in the current paper because the entire process is reasonably fast (less
than three hours), and we agree that it would better to briefly discuss it. It remains unknown that if StratLearner can
scale to large datasets (in terms of time and memory usage), and handling graphs with millions of nodes is nontrivial
because combinatorial algorithms are involved, which believe is a worthwhile research task.

**Reviewer 3**. **a)** This paper does not carry out the analysis of the final approximation guarantee on the PM problem, and
we would like to thank the reviewer for pointing out the concerns regarding this part, which we believe is an important
research direction. We are currently not able to establish the final approximation guarantee or the condition for ensuring
that the correct weight or distribution can be learned. The main challenge lies in relating the true error under the
structured prediction (with approximate inference) to the approximation ratio of the PM problem; there exist some
negative results for similar settings[2], showing that this problem is nontrivial. Nevertheless, the experimental results are
very promising, which is motivating us to continue theoretical explorations. **b)** Having observations from the cascades
is indeed a reasonable setting, but our problem is more concerned with the case in which we only have data regarding
the target problem (PM). We agree that it is interesting to examine the approaches that first learn the model and then
perform optimization, but we do not include them in the current experiments because they leverage another type of data.
**c)** MLP and GCN are trained using a standard classification loss. We will improve the description to make it clear.

**Reviewer 4.a)** The existing methods (e.g., Tong & Du [1]) require that the diffusion model is known to us, while our
paper deals with the case where the diffusion model is unknown and the entire problem has to be solved using data.
Therefore, the method in Tong & Du [1] is not applicable to our setting, and it is not used as a competitor. Because
the method in Tong & Du [1] gives the theoretically optimal solution (under the assumption NP$\neq$P), it is taken as a
baseline to evaluate StratLearner and its competitors. **b)** The number of the subgraphs determines the ability of our
model to approximate the underling diffusion function; therefore, the model trained with more subgraphs can produce a
better generalization performance.

We also thank the reviewers for other valuable comments: **a)** citing the related previous work right after Problem 1; **b)**
typo in line 99; **c)** clarifying that one cannot select existing seed for prevention; **d)** clarifying the NP-hardness in line
53; **e)** Line 70 is not correctly phrased. We will address them.

## Footnotes

[1]Leskovec, Jure, and Andrej Krevl. "SNAP datasets: stanford large network dataset collection; 2014."

[2]Balkanski, Eric, Aviad Rubinstein, and Yaron Singer. "The limitations of optimization from samples." STOC, 2017.


[Meta-Review · NeurIPS 2020]

The paper studies the problem of misinformation prevention in social networks in a novel setting where the underlying diffusion process of the network is unknown. The reviewers found the paper well-written and sound. The scalability to the methods to real-life sized social networks remains a challenge but the current work is a good advance.